# HawkesVAE: Sequential Patient Event Synthesis for Clinical Trials

## ABSTRACT

Generating sequential events data, such as adverse patient events, can provide valuable insights for clinical trial development, pharmaceutical research, patient modeling, and more. One approach to generate such data is by using generative AI models, which can synthesize data that resembles real-world data. However, in the domains such as clinical trials, patient data is especially limited. Data generation methods from literature such as LSTM, Probabilistic Auto-regressive, and Diffusion-based data generators struggle with this particular task off the shelf, as we show empirically. To address this task, we propose HawkesVAE, a Variational Autoencoder (VAE) that models events using Hawkes Processes (HP). Hawkes Processes are specialized statistical models designed specifically for the task of event and time-gap prediction, and VAEs enable an end-to-end generative design. Additionally, traditional VAEs rely solely on embeddings to decode events, but in a data-limited setting, this approach can have issues fitting the data. Therefore, we experiment with different ways of allowing the decoder to access varying amounts of information from the input events. Our experiments show that HawkesVAE outperforms other methods in terms of fidelity and allows the generation of highly accurate event sequences in multiple real-world sequential event datasets with only a small amount of external information. Furthermore, our empirical experiments demonstrate that HawkesVAE generates data that allows for superior utility over existing baselines while maintaining privacy.

## 1 INTRODUCTION

Generating sequential event data with limited training data is a crucial area of research, especially in healthcare (Wang & Sun, 2022a; Das et al., 2023; Theodorou et al., 2023). Clinical trials provide an example where generating synthetic event data for patients is particularly useful, as real data is often inaccessible due to patient privacy and legal reasons. Thus, high-quality synthetic sequential event data can be of great value for machine learning applications and data analysis. Our primary objective is to develop an effective algorithm for generating sequential event data with limited training data. However, developing a high-quality model for sequential data can be more complicated than data-rich tasks in computer vision or natural language processing. This is due to the diversity of individual features and the small training datasets available. Previous work in this area (Beigi et al., 2022; Shafquat et al., 2023) have generally focused on generating the static context information for each subject (e.g. subject demographics), while generating the sequential events has remained an elusive, yet vital next step in order to create a fully synthetic dataset.

Hawkes processes are statistical models that are specialized for event-type and time gap prediction (Hawkes, 1971), which has been shown to be highly effective at point process prediction when augmented with Transformer Layers (Zuo et al., 2020). Variational Autoencoders (VAEs) (Kingma & Welling, 2013) are a generative framework that specializes encoding an observed data into a probabilistic latent space $z$ that may be sampled. In our research, we demonstrate that combining the Hawkes Process with VAEs can successfully approximate sequential event generation, leading to state-of-the-art performance on our real-world data benchmarks.

To summarize our contributions,

1. We introduce `HawkesVAE`–a model that combines Variational Autoencoder + Hawkes Process that is able to generate sequential event data with their time gaps on 7 real-world clinical trial datasets. Additionally, our generation supports a high level of control, allowing users to specify specific event types to generate. To our knowledge, we are the first to propose this method for synthetic sequential event generation.

2. We demonstrate that `HawkesVAE` outperforms the alternative approaches in terms of AU-CROC for downstream ML classification designed for tabular data, including VAE+LSTM, PAR, and DDPM.

3. We conduct experiments demonstrating that `HawkesVAE`'s privacy with two metrics: ML Inference Score, which shows that synthetic event sequences are hard to distinguish from the original sequences, and Distance to Closest Record (DCR), which shows that synthetic sequences are not copies of the original data.

## 2 RELATED WORK

**Synthetic Data Generation** as a research area has been quickly garnering attention from the research community, with examples such as CTGAN (Xu et al., 2019), CTabGan (Zhao et al., 2022), TabDDPM (Kotelnikov et al., 2023), the Synthetic Data Vault[1] (Patki et al., 2016), and more. However, most of these models, such as TabDDPM and CTGAN, are focused on explicitly generating tabular data with no time component; or, in the case of SDV's ParSynthesizer (Zhang et al., 2022a), it is relatively simple and may be approximated with a GRU or LSTM model. In related fields such as synthetic clinical trial data generation (Choi et al., 2017; Das et al., 2023; Wang et al., 2023b; Lu et al., 2023; Theodorou et al., 2023), the model usually only focuses on generating the *order* at which certain clinical events happen (i.e., the diagnosis code of next patient visit), as opposed to generating the specific times of the visits as well. While it would be possible to extend this line of previous work, we believe that is out of scope for this paper and should be considered as future work.

**Hawkes Processes and VAEs** have been somewhat explored in the past. However, we found disadvantages that limit application in a full synthetic data generation setting. Previous work explores variational Hawkes processes in the context of event prediction for (disease progression (Sun et al., 2021) and social events sequences Pan et al. (2020), but they rely on the context of previous ground truth observations as well as the hidden state, whereas we can generate a synthetic sequence of event types and times. Another work (Lin et al., 2021) explores using variational approaches to disentangle multivariate Hawkes Process for event type prediction, but it also relies on knowing the ground truth to predict the next timestep. Additionally, our model only requires a single embedding to synthesize a sequence, whereas they require an embedding at each timestep. Furthermore, none of these methods are able solely to consider specific event types to generate, which `HawkesVAE` supports.

## 3 PROBLEM SETUP

### 3.1 NEURAL HAWKES PROCESS

We are given a set of $L$ observations of the form (time $t_j$, event_type $k_j$). $S = \{(t_1, k_1), \ldots, (t_j, k_j), \ldots, (t_L, k_L)\}$ Each time $t_j \in \mathbb{R}^+ \bigcup \{0\}$ and is sorted such that $t_j < t_{j+1}$. Each event $k_j \in \{1, \ldots, K\}$. The traditional Hawkes Process assumption that events only have a positive, decaying influence on future events is not realistic in practice, as there exist examples where an occurrence of an event lowers the probability of a future event (e.g., medication reduces the probability of adverse events). Therefore, the Neural Hawkes Process (Mei & Eisner, 2017) was proposed to generalize the traditional Hawkes Process. The following derivations follow (Zuo et al., 2020).

$$\lambda(t) := \sum_{k=1}^{K} \lambda_k(t) := \sum_{k=1}^{K} f_k(\boldsymbol{W}_k^\top \boldsymbol{h}(t)) = \sum_{k=1}^{K} \beta_k \log\left(1 + e^{\frac{\boldsymbol{W}_k^T \boldsymbol{h}(t)}{\beta_k}}\right),$$

---

[1] https://docs.sdv.dev/sdv/

where $\lambda(t)$ is the intensity function for *any* event occurring, $\lambda_k(t)$ is the intensity function for the event $k \in \mathcal{K}$ occurring, $K = |\mathcal{K}|$ is the total number of event types, and $h(t)$ are the hidden states of the event sequence obtained by a Transformer encoder and $\boldsymbol{W}_k^\top$ are learned weights that calculate the significance of each event type at time $t$. $f_k(c) = \beta_k \log(1 + e^{\frac{x}{\beta_k}})$ is the softplus function with parameter $\beta_k$. The output of $f_k(x)$ is always positive. Note that the positive intensity does not mean that the influence is always positive, as the influence of previous events are calculated through $\boldsymbol{W}_k^\top \boldsymbol{h}(t)$. If there is an event occurring at time $t$, then the probability of event $k$ is $P(k_t = k) = \frac{\lambda_k(t)}{\lambda(t)}$.

Let the history of all events before $t$ be represented by $\mathcal{H}_t = \{(t_j, k_j), t_j < t\}$. The continuous time intensity for prediction is defined as

$$\lambda(t|\mathcal{H}_t) := \sum_{k=1}^K \lambda_k(t|\mathcal{H}_t) := \sum_{k=1}^K f_k \left( \alpha_k \frac{t - t_j}{t_j} + \boldsymbol{W}_k^\top \boldsymbol{h}(t_j) + \mu_k \right),$$

where time is defined on interval $[t_j, t_{j+1})$, $f_k$ is the softplus function as before, $\alpha_k$ is a learned importance of the interpolation between the two observed timesteps $t_j$ and $t_{j+1}$. Note that when $t = t_j$, $\alpha_k$ does not matter as the influence is $0$ (intuitively, this is because we know that this event exists, so there is no need to estimate anything). The history of all previous events up to time $t$ is represented by $\boldsymbol{t}_j$. $\boldsymbol{W}_k^\top$ are weights that convert this history to a scalar. $\mu_k$ is the base intensity of event $k$. Therefore, the probability of $p(t|\mathcal{H}_{t_j})$ is the intensity at $t \in [t_j, t_{j+1})$ given the history $\mathcal{H}_t$ and the probability that no other events occur from the interval $(t_j, t)$

$$p(t|\mathcal{H}_{t_j}) = \lambda(t|\mathcal{H}_t) \exp \left( - \int_{t_j}^t \lambda(t'|\mathcal{H}_{t'}) dt' \right).$$

Note that if $t_j$ is the last observation, then $t_{j+1} = \infty$. Finally, the next time value $\hat{t}_{j+1}$ and event prediction $\hat{k}_{j+1}$ is given as

$$\hat{t}_{j+1} = \int_{t_j}^\infty t \cdot p(t|\mathcal{H}_t) dt, \quad \hat{k}_{j+1} = \arg\max_k \frac{\lambda_k(t_{j+1}|\mathcal{H}_{t_{j+1}})}{\lambda(t_{j+1}|\mathcal{H}_{t_{j+1}})}$$

For training, we want to maximize the likelihood of the observed sequence $\{(t_1, k_1), \ldots, (t_L, k_L)\}$. The log-likelihood function is given by[2]

$$\ell(\{(t_1, k_1), \ldots, (t_L, k_L)\}) = \sum_{j=1}^L \log(\lambda(t_j|\mathcal{H}_{t_j})) - \int_{t_1}^{t_L} \lambda(t|\mathcal{H}_t) dt.$$

Finally, since the gradient of the log-likelihood function has an intractable integral, one may obtain an unbiased estimate by performing Monte Carlo sampling (Robert et al., 1999).

$$\nabla \left[ \int_{t_1}^{t_L} \lambda(t|\mathcal{H}_t) dt \right]_{MC} = \sum_{j=2}^L (t_j - t_{j-1}) (\frac{1}{N} \sum_{i=1}^N \nabla \lambda(u_i))$$

With $u_i \sim Uniform(t_{j-1}, t_j)$. $\nabla \lambda(u_i)$ is fully differentiable with respect to $u_i$.

## 4 HAWKESVAE

Figure 1 shows an example of the proposed model with all optional structural constraints (allowing the model to access the true event knowledge, such as type and event length information). A diagram of the model without such additions is shown in Figure 3 in the Appendix. To combine the VAE and the Hawkes Process, we realize that the log-likelihood can be modeled as the log-likelihood of a Hawkes process if we assume that the event times and event types are generated from a multinomial Gaussian, i.e., the combined loss may be written as the following.

---

[2]The proof is shown in (Mei & Eisner, 2017) and Section A.8

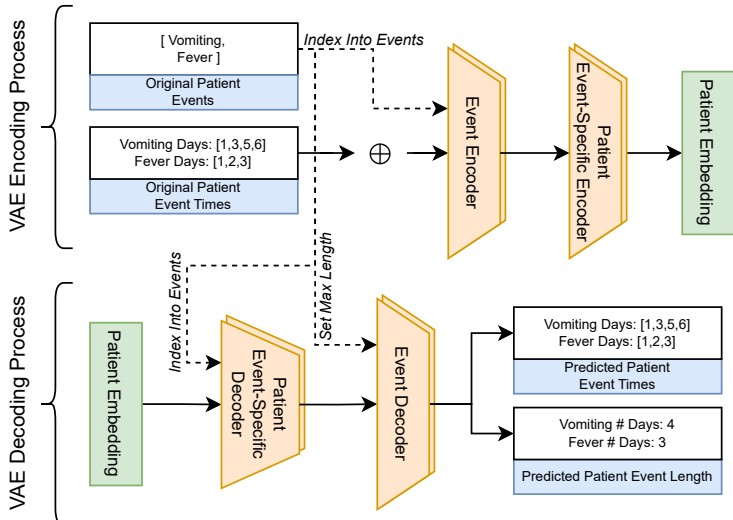

Figure 1: Diagram of the `HawkesVAE` Encoder-Decoder structure where. Here, the model input is the real patient event sequence + time, which is used to train a VAE model to the same output event sequence + time. The event sequence length for each event (if not known) is predicted as well. Optional inputs (*in dashed lines*) include the ground truth knowledge of which event types to generate and/or event lengths. The input observations are encoded first by a specific event encoder (Hawkes, LSTM, etc), and then embeddings are mapped to the patient embedding space via an MLP. The opposite process occurs for decoding from the patient embedding.

Sample event sequence $S_z \sim P_\theta(S|z)$ where $S_z = \{(t_1, k_1), \ldots, (t_j, k_j), \ldots, (t_{L_z}, k_{L_z})\}$. Then $H_{t,z}$ denotes the history up to time $t$ in $S_z$.

$$\lambda_\theta(t|\mathcal{H}_{t,z}) := \sum_{k=1}^{K} \lambda_{\theta,k}(t|\mathcal{H}_{t,z}) := \sum_{k=1}^{K} f_k\left(\alpha_k \frac{t - t_j}{t_j} + \boldsymbol{W}_{\theta,k}^\top \boldsymbol{h}_\theta(t_j) + \mu_{\theta,k}\right),$$

Where $t \in [t_j, t_{j+1})$. That is, $t$ lies between the $j$th and $j + 1$th observation in $S_z$ (if $t_j$ is the last observation, then $t_{j+1} = \infty$). $\lambda_{\theta,k}$, $\boldsymbol{W}_{\theta,k}^\top$, and $\boldsymbol{h}_\theta^\top$ are the same as the Neural Hawkes Process, only parameterized by $\theta$.

The log-likelihood is:

$$\ln P_\theta(S_z|z) = \sum_{j=1}^{L_z} \log(\lambda_\theta(t_j|\mathcal{H}_{t_j,z})) - \int_{t_1}^{t_{L_z}} \lambda_\theta(t|\mathcal{H}_{t,z})dt.$$

Adding the VAE ELBO loss (Appendix A.7), the combined `HawkesVAE` loss is:

$$L_{\theta,\phi} = \mathbb{E}_{z \sim q_\phi(\cdot|S_z)}\left[\ln P_\theta(S_z|z)\right] - D_{KL}(q_\phi(\cdot|S_z)||P_\theta(\cdot|S_z)).$$

## 4.1 KEY DETAILS

**Encoding and Decoding Hawkes Processes**    The encoder model $E_{\texttt{HawkesVAE}}(\mathcal{H}_i) \to \hat{\mu}, \hat{\sigma}$ takes in the original event types and times, and predicts the mean and standard deviation to sample $z \sim Normal(\hat{\mu}, \hat{\sigma})$. This is used for $q_\phi(\cdot|S_z)$ in the loss function and follows the previous Transformer Hawkes Process implementation of a transformer encoder, with alternating sine and cosine waves to denote temporal information as specified in Vaswani et al. (2017).

However, the decoder model is more complicated. Hawkes Process is usually evaluated via one prediction step at a time (i.e., the input is the ground truth time-step and event type, and the task is to predict the next time step and event type). For our purposes, we need to adapt this into an autoregressive decoding scheme similar to the language modeling task.

At *training time*, the input to the decoder $D_{\texttt{HawkesVAE}}(z, \mathcal{H}_i) \rightarrow (\hat{t}_{i+1}, \hat{k}_{i+1}, \lambda)$ is hidden vector $z$ and a sequence of ground truth event types and times. It is tasked with predicting the next event type $\hat{k}$, next event time $\hat{t}$, and the $\lambda$s necessary to compute $P_\theta(\cdot|S_z)$. Furthermore, we follow Transformer Hawkes Process's approach of also adding mean squared error losses to the time: $time\_loss = \|t - \hat{t}\|^2$ and cross-entropy loss of the predicted $type\_loss = -\sum_{c=1}^{|\mathcal{K}|} k \log(p_k)$.

At *inference time*, the input to decoder is only $z$, and we auto-regressively decode the predicted event types and times. To predict next time and event tuple $(\hat{t}_i, \hat{k}_i)$, the input is the previously predicted times and events $\{(\hat{t}_1, \hat{k}_1), \ldots, (\hat{t}_{i-1}, \hat{k}_{i-1}))\})$. (each predicted time and event is repeatedly appended to the input). We find that auto-regressive greedy sampling $\hat{k} = \arg\max(\hat{\boldsymbol{k}})$ of the next event type does not result in good sequences, and probabilistic sampling based on the raw logit score will result in higher utility sequences. Finally, we note that we can control for generating events that are similar to the original patient by first encoding the original patient, and then sampling around it, a benefit of the probabilistic nature of the VAE latent space $z$. Otherwise, it would be impossible to correspond the original labels to the synthetic data.

**Event Type Information**    In addition to proposing `HawkesVAE`, we also propose several variants of it. In some applications, such as clinical trial patient modelling (Wang & Sun, 2022a; Fu et al., 2022; Das et al., 2023), we may be interested in an event sequence *with the event types known*, that is, the model only needs to generate the timestamps at which events occur. This is to address the concern of subject fidelity–that is–the generated subject must be significantly similar to the original subject in order for the generated data to be useful; therefore, knowing which events occur in a subject to generate a similar subject would not be unreasonable. The "Events Known" model was created to enforce ONLY simulating specific events, without consideration of all events (which may be irrelevant and confuse the generator).

To accommodate this framework, we train $num\_event$ Transformer Hawkes Process expert encoders and decoders that only model a single event. Since event type is known, we may index into the specific encoder/decoder pair as given by the event index (shown in Figure 1). Finally, all independent event time predictions over all known types are combined via their predicted event times to create a final multi-event sequence prediction. Since the VAE model requires a single embedding to sample a latent vector, we sum the patient embedding output of all expert event encoders, and pass this joint embedding to the expert decoders. The decoder is trained to generate the predicted time and type sequence of its respective expert event.

Note that we have the sum the latent patient embeddings of the expert decoders due to the limitation of having a fully generative model. It would be unfair for to have varying size latent spaces for each patient.

**Sequence Length Prediction**    Given known subset of events $\mathcal{K}' \in \mathcal{K}$, we generate event sequences such that $S = \{(t_j, k_j); j = 1, \ldots, L; k_j \in \mathcal{K}'\}$, where $L$ is learned from the data. *Note that by default, all `HawkesVAE` variants autoregressively continue to predict event types and times until it reaches its own predicted stopping length.* Like other synthesizers (Zhang et al., 2022a), we also experiment with using a max-length criterion, such that the event generation still stops at a certain $L$. We explore 2 cases. (1) If the event type *is not known*, we simply generate sequence $S = \{(t_j, k_j); j = 1, \ldots, L'\}$ where $L'$ is a prespecified sequence length. (2) If the event type *is known*, then we allow the model to know the specific max lengths $L'_k$ for each specific event type. Let $L' = \sum_{k_i \in \mathcal{K}'} L'_{k_i}$ We then generate a sequence $S = \{(t_j, k_j); j = 1, \ldots, L'; \left(\sum_{j=1}^{L'} \mathbb{1}(\hat{k}_j = k_i)\right) = L'_{k_i}; \forall k_i \in \mathcal{K}'\}$

## 5    EXPERIMENTS

**Datasets**    We evaluated our models on 7 real-world clinical trial outcome datasets obtained from Project Data Sphere[3] (Green et al., 2015; Fu et al., 2022). Specifically, we chose the trials as outlined in Table 1. These datasets have shown to be effective evaluation datasets for tabular prediction (Wang & Sun, 2022b; Wang et al., 2023a) and digital twin generation (Das et al., 2023). Specifically, we use

---

[3]https://data.projectdatasphere.org/projectdatasphere/html/access

Table 1: A description of all of the real-world datasets used in the evaluation. All trial data was obtained from Project Data Sphere (Green et al., 2015). Num Rows refers to the raw number of data points in the trial. Num Subj refers to the total number of patients. Num Events denotes the total number of *unique* events. Events / Subj denotes the average number of events that a patient experiences. Positive Label Proportion denotes the percentage of patients that did not experience the death event.

| Dataset | Description | Num Rows | Num Subj | Num Events | Events / Subj | Positive Label Proportion |
|---|---|---|---|---|---|---|
| NCT00003299 | Small Cell Lung Cancer | 20210 | 548 | 34 | 36.880 | 0.951 |
| NCT00041119 | Breast Cancer | 2983 | 425 | 150 | 7.019 | 0.134 |
| NCT00079274 | Colon Cancer | 316 | 70 | 18 | 4.514 | 0.184 |
| NCT00174655 | Breast Cancer | 7002 | 953 | 21 | 7.347 | 0.019 |
| NCT00312208 | Breast Cancer | 2193 | 378 | 182 | 5.802 | 0.184 |
| NCT00694382 | Venous Thromboembolism in Cancer Patients | 7853 | 803 | 746 | 9.780 | 0.456 |
| NCT03041311 | Small Cell Lung Cancer | 1043 | 47 | 207 | 22.192 | 0.622 |

NCT00003299 (Niell et al., 2005), NCT00041119 (Baldwin et al., 2012), NCT00079274 (Alberts et al., 2012), NCT00174655 (Fernández-Cuesta et al., 2012), NCT00312208 (Mackey et al., 2016), NCT00694382 (Agnelli et al., 2012), NCT03041311 (Daniel et al., 2021). A full description of the data is shown in Table 1. Each dataset contains events and the times at which they occur, e.g. medications, procedures, as well as some adverse events like vomiting, etc. We use these datasets to predict if the subject experiences the death event, which is an external label. Note that `HawkesVAE` does not require a fixed patient event sequence length.

**Models to Compare**     We compared the following models:

**1. LSTM VAE**: To compare against a VAE baseline, we manually implement our own LSTM VAE, which predicts the event type as a categorical classification task and the timestamp as a regression task at each event prediction.

**2. PARSynthesizer** from SDV (Zhang et al., 2022a; Patki et al., 2016) since it is the most relevant model for synthesizing sequential event data, based on a conditional probabilistic auto-regressive (CPAR) model. To the best of our knowledge, no other models specifically handle sequential event data generation from scratch with easily accessible code.

**3. DDPM** (Kotelnikov et al., 2023) is a recently proposed state-of-the-art general tabular synthesizer based on diffusion models. Although it is not explicitly built for sequential data, we are able to enhance it by adding time as a numerical column. This model also outperforms CTGAN-related models Xu et al. (2019); Zhao et al. (2021; 2022), the previous go-to for synthetic tabular data generation. We believe that this is a strong, representative baseline of general tabular synthetic data generation.

**3. `HawkesVAE` (Multivariate)** is the VAE + Multivariate Hawkes Process that is trained without any assumptions. At training time, the task is to predict the next event and timestamp given a history of observations. At inference time, we perform autoregressive decoding, adding the last prediction to the history until a predicted cutoff is reached.

**4. `HawkesVAE` (Events Known)** assumes that one knows which specific events occur for the Hawkes Model (but not the timestamps or the number of occurrences). Similar to `HawkesVAE` (Multivariate), training is performed by predicting the next event and timestamp given history, and autoregressive decoding is done at inference time until a predicted cutoff.

Additionally, we attempted to implement HALO (Theodorou et al., 2023), a hierarchical autoregressive language model that achieved state-of-the-art performance for Electronic Health Record (EHR)

synthesis, but was not able to obtain representative results on the clinical trial evaluation datasets, primarily due to the small size of training data, demonstrating the difficulty of this task.

## 5.1 UTILITY EVALUATION

We evaluate the utility (ROCAUC) of the generated synthetic data by performing binary classification of death events in all 7 clinical trials. The standard deviation of each ROCAUC score is calculated via bootstrapping (100x bootstrapped test datapoints). Training is performed completely on synthetic data by matching each generated patient to its ground truth death event label. Testing is performed on the original held-out ground truth split. For the Original Data baseline, we performed 5 cross validations on 80/20 train test splits of the real data. The main results are shown in Table 2. We see that synthetic data generated by HawkesVAE variants generally perform the best in terms of downstream death event classification performance, where HawkesVAE (Multivariate) outperforms the next best model (in 4/7 datasets and is within 1 standard deviation with the rest of the datasets). Allowing the model access to the specific types also enables it to significantly outperform other baselines. Occasionally, synthetic data is able to support better performance than the original dataset on downstream tasks (this behavior is also seen in TabDDPM). We believe that this is due to the synthetic model generating examples that are more easily separable and/or more diverse than real data. However, this is only a hypothesis and should be investigated further in future research, but we are encouraged to see that our proposed method captures this behavior. Additionally, some examples of an (anonymized) generated sequence can be seen in Figure 2.

Table 2: Binary classification ROCAUCs (↑ higher the better, ± standard deviation) of a downstream LSTM trained on data generated from the HawkesVAE models as well as the original data and baselines. Note that the LSTM and the HawkesVAE models estimate their own sequence length. HawkesVAE (Events Known) is put in a separate category due to its requirement of event type information. **Bolded** indicates original data mean within 1 standard deviation.

| Dataset | Original Data | LSTM VAE | PAR | DDPM | Hawkes (Multivariate) | Hawkes (Events Known) |
|---|---|---|---|---|---|---|
| NCT00003299 | $0.689 \pm 0.105$ | $0.563 \pm 0.053$ | $0.504 \pm 0.066$ | $0.557 \pm 0.055$ | $0.572 \pm 0.051$ | $\mathbf{0.709 \pm 0.049}$ |
| NCT00041119 | $0.678 \pm 0.078$ | $0.617 \pm 0.036$ | $0.573 \pm 0.043$ | $\mathbf{0.630 \pm 0.045}$ | $\mathbf{0.646 \pm 0.037}$ | $\mathbf{0.665 \pm 0.045}$ |
| NCT00079274 | $0.637 \pm 0.140$ | $0.481 \pm 0.092$ | $\mathbf{0.567 \pm 0.096}$ | $\mathbf{0.583 \pm 0.098}$ | $\mathbf{0.622 \pm 0.016}$ | $\mathbf{0.653 \pm 0.019}$ |
| NCT00174655 | $0.660 \pm 0.128$ | $0.535 \pm 0.073$ | $0.523 \pm 0.074$ | $0.513 \pm 0.078$ | $0.548 \pm 0.024$ | $\mathbf{0.594 \pm 0.068}$ |
| NCT00312208 | $0.632 \pm 0.072$ | $0.454 \pm 0.039$ | $0.463 \pm 0.039$ | $0.503 \pm 0.043$ | $\mathbf{0.590 \pm 0.050}$ | $\mathbf{0.634 \pm 0.032}$ |
| NCT00694382 | $0.640 \pm 0.038$ | $0.490 \pm 0.019$ | $0.549 \pm 0.022$ | $0.531 \pm 0.021$ | $0.568 \pm 0.018$ | $\mathbf{0.625 \pm 0.020}$ |
| NCT03041311 | $0.738 \pm 0.149$ | $0.563 \pm 0.097$ | $0.507 \pm 0.087$ | $0.574 \pm 0.096$ | $\mathbf{0.689 \pm 0.084}$ | $\mathbf{0.755 \pm 0.059}$ |

## 5.2 PRIVACY EVALUATIONS

**ML Inference Score**: We first calculate the performance of predicting whether a generated sequence is real vs synthetic via an LSTM binary classification (Patki et al., 2016). The real subjects are labelled with "0" and the synthetic subjects are labelled with "1". Results are shown in Table 3, and we see that HawkesVAE variants perform closest to the optimal 0.5 ROCAUC ideal score. One thing to note is that a perfect copy of the original data would result in a 0.5 score, so we have the following metric to measure the opposite scenario.

**Distance to Closest Record (DCR) Score**: Second, we follow the evaluation metrics per TabDDPM (Kotelnikov et al., 2023). That is, we compare the feature vectors of the real vs synthetic data, and measure how far the synthetic data is from the original. The higher this distance is, the more different the generated data is from the original data, and thus the more private it is. A completely different version of the data would obtain the highest distance, but could result in bad performance in the downstream LSTM classification performance or a high ML Inference score (close to 1). We calculate this by featurizing the event time predictions in terms of (count, means, and standard deviations). Then, we normalize and obtain the L2 distance between a generated subject and the closest real subject. Table 4 shows this result. Notice that HawkesVAE variants generally obtain quite low scores on this metric. DDPM and PAR also generate data closer to the original data compared to LSTM VAE. We note the privacy-fidelity tradeoff, as LSTM VAE generates data that is further away from the original, but yields worse utility (Table 2).

Table 3: Results of ML Inference Score: LSTM binary classification of real vs synthetic (*the closer to 0.5 the score is, the better*). Standard deviation calculated via bootstrapping is shown via $\pm$. AUCROC scores are shown. `HawkesVAE` (Events Known) is put in a separate category due to its requirement of event type info.

| Dataaest | LSTM VAE | PAR | DDPM | HawkesVAE (Multivariate) | HawkesVAE (Events Known) |
|---|---|---|---|---|---|
| NCT00003299 | $1.000 \pm 0.000$ | $0.968 \pm 0.010$ | $0.762 \pm 0.024$ | $0.792 \pm 0.019$ | $0.689 \pm 0.020$ |
| NCT00041119 | $0.932 \pm 0.017$ | $0.998 \pm 0.002$ | $0.926 \pm 0.017$ | $0.726 \pm 0.015$ | $0.768 \pm 0.021$ |
| NCT00079274 | $1.000 \pm 0.000$ | $0.807 \pm 0.082$ | $0.894 \pm 0.050$ | $0.733 \pm 0.012$ | $0.701 \pm 0.054$ |
| NCT00174655 | $1.000 \pm 0.000$ | $0.999 \pm 0.001$ | $0.998 \pm 0.001$ | $0.696 \pm 0.008$ | $0.593 \pm 0.023$ |
| NCT00312208 | $0.994 \pm 0.007$ | $0.874 \pm 0.026$ | $0.729 \pm 0.035$ | $0.712 \pm 0.024$ | $0.693 \pm 0.038$ |
| NCT00694382 | $1.000 \pm 0.000$ | $0.923 \pm 0.012$ | $0.992 \pm 0.005$ | $0.891 \pm 0.014$ | $0.856 \pm 0.016$ |
| NCT03041311 | $1.000 \pm 0.000$ | $0.651 \pm 0.112$ | $0.374 \pm 0.121$ | $0.573 \pm 0.111$ | $0.477 \pm 0.127$ |

Table 4: Distance to Closest Record (DCR) Score. Note that this score only tells part of the picture. The higher this score is, the larger the difference between the synthetic data and the original data. The lower the score, the more similar the synthetic data is to the original data.

| Dataset | LSTM VAE | PAR | DDPM | HawkesVAE (Multivariate) | HawkesVAE (Events Known) |
|---|---|---|---|---|---|
| NCT00003299 | 3.700 | 2.647 | 1.426 | 2.256 | 1.138 |
| NCT00041119 | 4.677 | 4.633 | 1.007 | 3.251 | 0.612 |
| NCT00079274 | 2.732 | 1.977 | 1.346 | 1.618 | 1.675 |
| NCT00174655 | 32.185 | 56.915 | 3.581 | 2.110 | 1.215 |
| NCT00312208 | 87.015 | 2.348 | 1.207 | 1.535 | 0.745 |
| NCT00694382 | 17.946 | 35.362 | 1.059 | 2.125 | 0.971 |
| NCT03041311 | 36.740 | 37.723 | 4.662 | 5.565 | 4.922 |

## 5.3 ABLATIONS

**Assuming Knowledge of Event Lengths**   In this section, we examine the ability of our framework to take in additional subject-level information regarding sequence generation. For example, `HawkesVAE` assumes that we have access to the original event lengths (e.g., event 1 occurs 5 times in the original sequence, generate the times at which event 1 occurs), as well as the event indices (e.g., we know that subject 1 has events 2,5,6). For more details, see Section 4.1. Table 5 shows the result of allowing the model to know how many times events occur (for `HawkesVAE` (Events Known)) or how the total length of the sequence to be generated (for `HawkesVAE` (Multivariate) and LSTM VAE). We see that providing the model with a list of event types that occur boosts performance significantly, as `HawkesVAE` (Events Known) performs markedly better than the other models.

Table 5: AUCROC results ($\uparrow$ higher the better, $\pm$ standard deviation) from fitting a downstream LSTM to predict death event from the ablation of informing of the exact length of the sequence to generate.

| Dataset | Original Data | LSTM | Hawkes (Multivariate) | Hawkes (Events Known) |
|---|---|---|---|---|
| NCT00003299 | $0.689 \pm 0.105$ | $0.485 \pm 0.050$ | $0.574 \pm 0.071$ | $0.761 \pm 0.038$ |
| NCT00041119 | $0.678 \pm 0.078$ | $0.613 \pm 0.037$ | $0.643 \pm 0.039$ | $0.652 \pm 0.042$ |
| NCT00079274 | $0.637 \pm 0.140$ | $0.403 \pm 0.094$ | $0.632 \pm 0.017$ | $0.654 \pm 0.098$ |
| NCT00174655 | $0.670 \pm 0.128$ | $0.560 \pm 0.053$ | $0.557 \pm 0.057$ | $0.618 \pm 0.053$ |
| NCT00312208 | $0.632 \pm 0.072$ | $0.437 \pm 0.040$ | $0.589 \pm 0.041$ | $0.624 \pm 0.037$ |
| NCT00694382 | $0.640 \pm 0.038$ | $0.496 \pm 0.021$ | $0.585 \pm 0.022$ | $0.642 \pm 0.022$ |
| NCT03041311 | $0.738 \pm 0.149$ | $0.608 \pm 0.092$ | $0.717 \pm 0.083$ | $0.860 \pm 0.056$ |

**HawkesVAE for Event Forecasting**   The results in Table 6 show that the `HawkesVAE` variants are generally significantly better than their other counterparts in terms of event forecasting. Note that it is possible for `HawkesVAE` (Events Known) not to have perfect accuracy in terms of event prediction because the time prediction may be incorrect, and therefore, the predicted ordering of the events could not match the original ordering.

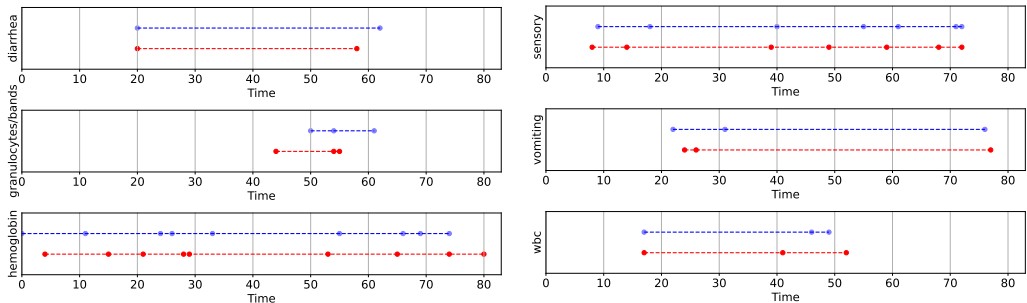

Figure 2: Example of a generated sequence from `HawkesVAE` (Events Known) from NCT00003299 plotted by the individual events. The blue dots denoting the specific event timestamp prediction. The red dots are the ground truth timestamps and the ground truth predictions. Each prediction is linked with dashed lines for clarity.

Table 6: Accuracy of `HawkesVAE` for Event Forecasting ($\uparrow$ higher the better, $\pm$ standard deviation). A correct prediction is made if raw predicted event matches the original event in the ordering respectively. Note that `HawkesVAE` (Events Known) is a special case since it only needs to generate event types from a given event list.

| Dataset | LSTM VAE | PAR | DDPM | HawkesVAE (Multivarate) | HawkesVAE (Events Known) |
|---|---|---|---|---|---|
| NCT00003299 | $0.043 \pm 0.047$ | $0.052 \pm 0.048$ | $0.056 \pm 0.054$ | $0.023 \pm 0.034$ | $0.602 \pm 0.085$ |
| NCT00041119 | $0.001 \pm 0.013$ | $0.110 \pm 0.151$ | $0.339 \pm 0.283$ | $0.357 \pm 0.012$ | $0.848 \pm 0.228$ |
| NCT00079274 | $0.028 \pm 0.074$ | $0.073 \pm 0.145$ | $0.376 \pm 0.255$ | $0.406 \pm 0.101$ | $0.765 \pm 0.234$ |
| NCT00174655 | $0.134 \pm 0.062$ | $0.093 \pm 0.144$ | $0.117 \pm 0.005$ | $0.158 \pm 0.340$ | $0.992 \pm 0.055$ |
| NCT00312208 | $0.002 \pm 0.022$ | $0.037 \pm 0.112$ | $0.154 \pm 0.225$ | $0.102 \pm 0.031$ | $0.530 \pm 0.294$ |
| NCT00694382 | $0.000 \pm 0.005$ | $0.015 \pm 0.053$ | $0.033 \pm 0.168$ | $0.051 \pm 0.009$ | $0.360 \pm 0.267$ |
| NCT03041311 | $0.000 \pm 0.005$ | $0.036 \pm 0.051$ | $0.130 \pm 0.133$ | $0.140 \pm 0.013$ | $0.265 \pm 0.215$ |

## 6 DISCUSSION

Creating sequential events and data of significance is crucial for advancing clinical trial development, pharmaceutical research, and related domains. However, in many of these fields, strict legal privacy regulations have resulted in much of this valuable data being isolated and inaccessible. Generation of synthetic data addresses this, but sparse event occurrences and limited training data increase the complexity and difficulty of this task.

We introduce `HawkesVAE`, a novel model that combines Variational Autoencoder and Hawkes Process techniques. While we have evaluated `HawkesVAE` on clinical trial patient data, the methodology is quite general and can be applied to other forms of sequential data, such as financial or social media data. This model excels at generating sequential event data with precise timestamps, and we demonstrate its superior performance compared to the traditional LSTM and GAN-based models in handling tabular data for downstream machine learning tasks. Additionally, `HawkesVAE` showcases its ability to forecast events. Through experiments, we illustrate its capacity to generate event sequences that closely resemble the originals, without resorting to mere duplication. Ultimately, we demonstrate that `HawkesVAE` outperforms existing methods in terms of data utility, enabling the generation of highly authentic event sequences across multiple real-world sequential event datasets. Empirical experiments indicate that providing the model with additional information, such as event index or event length, leads to significant improvements in the synthetic data quality. We believe that a sweet spot is reached by allowing the model to know the event index–as it provides a significant downstream classification boost while maintaining a low ML inference score.

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

**Contents**

# A APPENDIX

## A.1 ETHICS AND REPRODUCIBILITY

Transformer Hawkes (Zuo et al., 2020) is open source and can be found at `https://github.com/SimiaoZuo/Transformer-Hawkes-Process`. Training on an NVIDIA GeForce RTX 3090 takes around 12 hrs to run the full model. The code will be made public and open source on GitHub. for the camera-ready version. All datasets were obtained from Project Data Sphere (Green et al., 2015) with permission via a research data access request form. The links are as follows:

1. NCT00003299 (Niell et al., 2005): A Randomized Phase III Study Comparing Etoposide and Cisplatin With Etoposide, Cisplatin and Paclitaxel in Patients With Extensive Small Cell Lung Cancer. Available at `https://data.projectdatasphere.org/projectdatasphere/html/content/261`

2. NCT00041119 (Baldwin et al., 2012): Cyclophosphamide And Doxorubicin (CA) (4 VS 6 Cycles) Versus Paclitaxel (4 VS 6 Cycles) As Adjuvant Therapy For Breast Cancer in Women With 0-3 Positive Axillary Lymph Nodes:A 2X2 Factorial Phase III Randomized Study. Available at `https://data.projectdatasphere.org/projectdatasphere/html/content/486`

3. NCT00079274 (Alberts et al., 2012): A Randomized Phase III Trial of Oxaliplatin (OXAL) Plus 5-Fluorouracil (5-FU)/Leucovorin (CF) With or Without Cetuximab (C225) After Curative Resection for Patients With Stage III Colon Cancer. Available at `https://data.projectdatasphere.org/projectdatasphere/html/content/407`

4. NCT00174655 (Fernández-Cuesta et al., 2012): An Intergroup Phase III Trial to Evaluate the Activity of Docetaxel, Given Either Sequentially or in Combination With Doxorubicin, Followed by CMF, in Comparison to Doxorubicin Alone or in Combination With Cyclophosphamide, Followed by CMF, in the Adjuvant Treatment of Node-positive Breast Cancer Patients. Available at `https://data.projectdatasphere.org/projectdatasphere/html/content/127`

5. NCT00312208 (Mackey et al., 2016): A Multicenter Phase III Randomized Trial Comparing Docetaxel in Combination With Doxorubicin and Cyclophosphamide Versus Doxorubicin and Cyclophosphamide Followed by Docetaxel as Adjuvant Treatment of Operable Breast Cancer HER2neu Negative Patients With Positive Axillary Lymph Nodes. Available at `https://data.projectdatasphere.org/projectdatasphere/html/content/118`

6. NCT00694382 (Agnelli et al., 2012): A Multinational, Randomized, Double-Blind, Placebo-controlled Study to Evaluate the Efficacy and Safety of AVE5026 in the Prevention of Venous Thromboembolism (VTE) in Cancer Patients at High Risk for VTE and Who Are Undergoing Chemotherapy. Available at https://data.projectdatasphere.org/projectdatasphere/html/content/119

7. NCT03041311 (Daniel et al., 2021): Phase 2 Study of Carboplatin, Etoposide, and Atezolizumab With or Without Trilaciclib in Patients With Untreated Extensive-Stage Small Cell Lung Cancer (SCLC). Available at https://data.projectdatasphere.org/projectdatasphere/html/content/435

### A.1.1 HAWKESVAE HYPERPARAMETERS

For PARSyntheizer, default hyper-parameters were used. For DDPM, we followed the GitHub example for churn2 https://github.com/yandex-research/tab-ddpm, but trained for 10,000 steps for each dataset.

Table 7: Hyperparameters Considered for HawkesVAE

| Parameter | Space |
| --- | --- |
| embedding_size | [32,64,128] |
| patient_embedding_size | [64,128,256] |
| num_transformer_layers (Encoder) | [1,2,3,4,5,6,7,8] |
| num_heads (Encoder) | [2,4,8] |
| num_transformer_layers (Decoder) | [1,2,3,4,5,6,7,8] |
| num_heads (Decoder) | [2,4,8] |
| lr | [1e-3, 1e-4] |

Table 8: Hyperparameters Considered for LSTM VAE

| Parameter | Space |
| --- | --- |
| embedding_size | [32,64,128] |
| patient_embedding_size | [64,128,256] |
| num_lstm_layers (Encoder) | [1,2] |
| hidden_size (Encoder) | [32,64,128] |
| num_lstm_layers (Decoder) | [1,2] |
| hidden_size (Decoder) | [32,64,128] |
| lr | [1e-3, 1e-4] |

### A.1.2 ML UTILITY CALCULATION HYPERPARAMETERS

This section outlines hyperparameters explored for the downstream model for downstream ML Utility.

Table 9: Hyperparameters Considered for LSTM Predictor Models

| Parameter | Space |
| --- | --- |
| embedding_size | [32,64,128] |
| num_lstm_layers (Encoder) | [1,2] |
| hidden_size (Encoder) | [32,64,128] |
| lr | [1e-3, 1e-4] |

### A.2 MORE RELATED WORK

**Hawkes Processes** (Hawkes, 1971; Isham & Westcott, 1979; Liniger, 2009) are point process that models event occurrence times as a function of previous event occurrences. Recent work has gener-

alized this classical model to achieve state-of-the-art results in point process modeling tasks, such as estimating the causal influence of misinformation on social media (Zhang et al., 2022b), knowledge graph temporal entity/time prediction (Choi et al., 2015; Fu et al., 2021; Sun et al., 2022), neural differential equations (Chen et al., 2018; Kidger et al., 2020), time series prediction (Wen et al., 2022), and more.

**Variational Autoencoders (VAEs)** (Kingma & Welling, 2013) are generative models that have been applied to many data-synthesizing tasks such as probabilistic multivariate time series imputation (Fortuin et al., 2020) or generating diverse videos from a single input example video (Gur et al., 2020). Furthermore, VAEs have the unique advantage of being able to sample from the embedding distribution. Therefore, at inference time, we have the option to sample around the encoded embeddings of a certain data point.

### A.3 AUTOREGRESSIVE GREEDY SAMPLING VS PROBABILISTIC SAMPLING

In this section, we compare the results of greedily choosing the best type as opposed to probabilistic sampling from the rescaled logits $l \in \mathbb{R}^{N_{events}}$. We rescale the logits to be positive and sum to 1: $l = \frac{l - min(l)}{\sum l - min(l)}$, and sample from $k \sim Multinomial(l)$ (For the Greedy version, we take $\arg\max_k(l)$).

Let `HawkesVAE` (Probabilistic) represent the probabilistic sampling and `HawkesVAE` (Greedy) represent greedy sampling. Table 10 shows the results, and we see that the Greedy version often fails to generate useful synthetic sequences.

Table 10: Binary death event classification ROCAUCs of a downstream LSTM trained on data generated from the `HawkesVAE` models as well as the original data and baselines. Note that the LSTM and the `HawkesVAE` models estimate their own sequence stopping length.

| Dataset | NCT00003299 | NCT00041119 | NCT00079274 | NCT00174655 |
|---|---|---|---|---|
| Original Data | $0.5608 \pm 0.0891$ | $0.6322 \pm 0.0679$ | $0.6009 \pm 0.1525$ | $0.6008 \pm 0.1015$ |
| HawkesVAE (Probabalistic) | $0.5860 \pm 0.0564$ | $0.6379 \pm 0.0378$ | $0.6007 \pm 0.0831$ | $0.5775 \pm 0.0538$ |
| HawkesVAE (Greedy) | $0.4525 \pm 0.0517$ | $0.6287 \pm 0.0420$ | $0.3909 \pm 0.0939$ | $0.4505 \pm 0.0919$ |

| Dataset | NCT00312208 | NCT00694382 | NCT03041311 |
|---|---|---|---|
| Original Data | $0.5285 \pm 0.0566$ | $0.5915 \pm 0.0281$ | $0.6046 \pm 0.1244$ |
| HawkesVAE (Probabilistic) | $0.5632 \pm 0.0384$ | $0.5149 \pm 0.0198$ | $0.5584 \pm 0.0741$ |
| HawkesVAE (Greedy) | $0.4495 \pm 0.0142$ | $0.3829 \pm 0.0613$ | $0.5001 \pm 0.0341$ |

### A.4 DIAGRAM OF THE VAE ENCODER-DECODER STRUCTURE USING A SINGLE MODEL

In the main paper, we show the diagram of `HawkesVAE` (Events Known). For VAE LSTM and `HawkesVAE` (Multivariate), we do not need the additional complexity of learning separate weights for each event, as we simply model all events in a multivariate fashion. Figure 3 shows this construction.

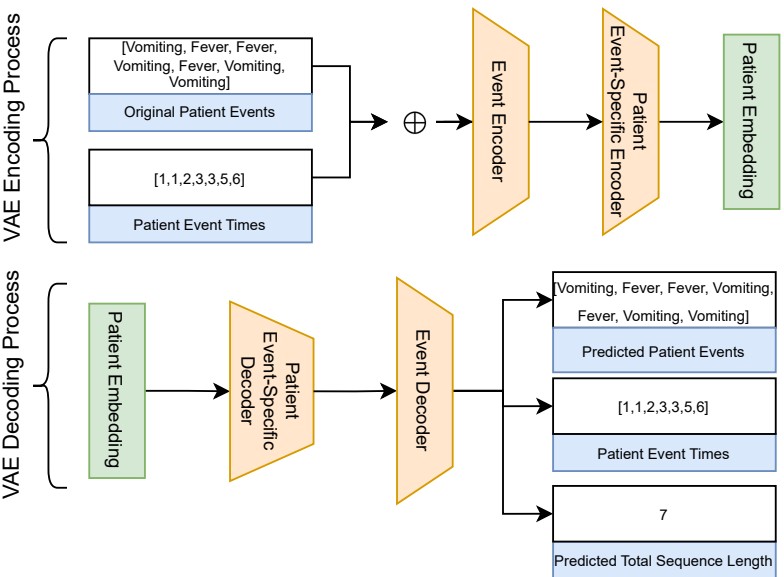

Figure 3: Diagram of `HawkesVAE` (Multivariate) Encoder-Decoder structure, where event types and lens are learned. Here, the model output is the event times, event types, and total sequence length. The input observations are encoded first by a specific event encoder (Hawkes, LSTM, etc), and then embeddings are then mapped to the patient embedding space via an MLP. The opposite process occurs for decoding from the patient embedding.

## A.5 PLOTS OF SUBJECTS

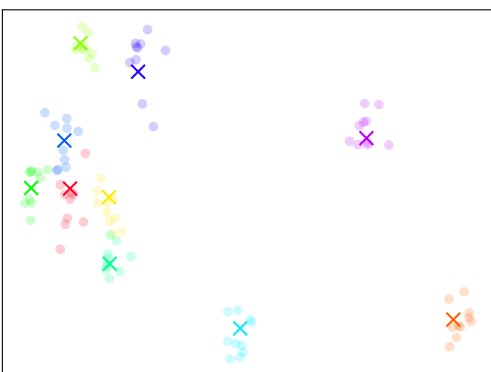

Figure 4: Plot of 10 subject embeddings (in different colors) with 10 randomly sampled embeddings surrounding them. Embeddings are obtained from the predicted mean and standard deviation. The original predicted mean is shown with a $\times$ and the sampled embeddings are shown with a dot. These embeddings can be used to generate event sequences that should look somewhat similar to the original.

In this section, we visualize some of the `HawkesVAE` embeddings. Using Principal Component Analysis (PCA) to obtain a 2-dimensional visualization, we show 10 randomly sampled embeddings surrounding an initial subject embedding. The VAE loss penalizes event sequences generated from these similar embeddings to be the same as the original patient embedding. Figure 4 shows this plot,

which makes sense as a sanity check that the sampled subject embeddings are close to the original and generally do not overlap with other subjects.

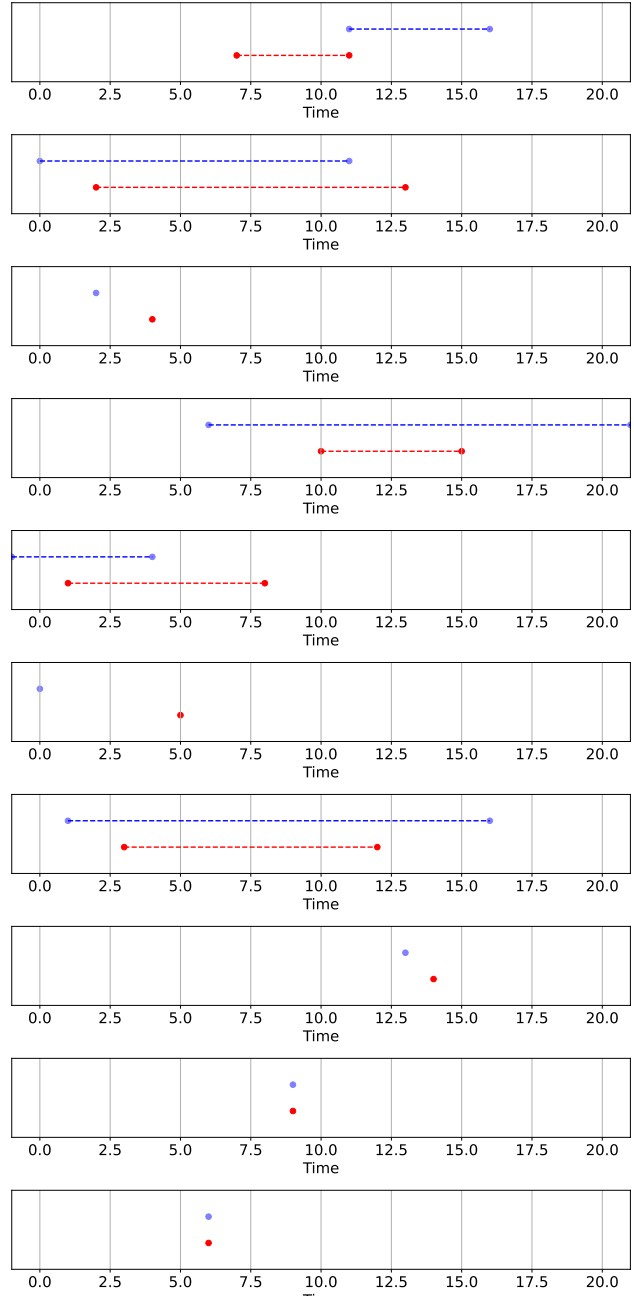

Figure 5: Example of another generated sequence from HawkesVAE (Events Known) from NCT00003299. The blue dots denoting the specific event timestamp prediction. The red dots are the ground truth timestamps and the ground truth predictions. Each prediction is also linked with dashed lines for clarity

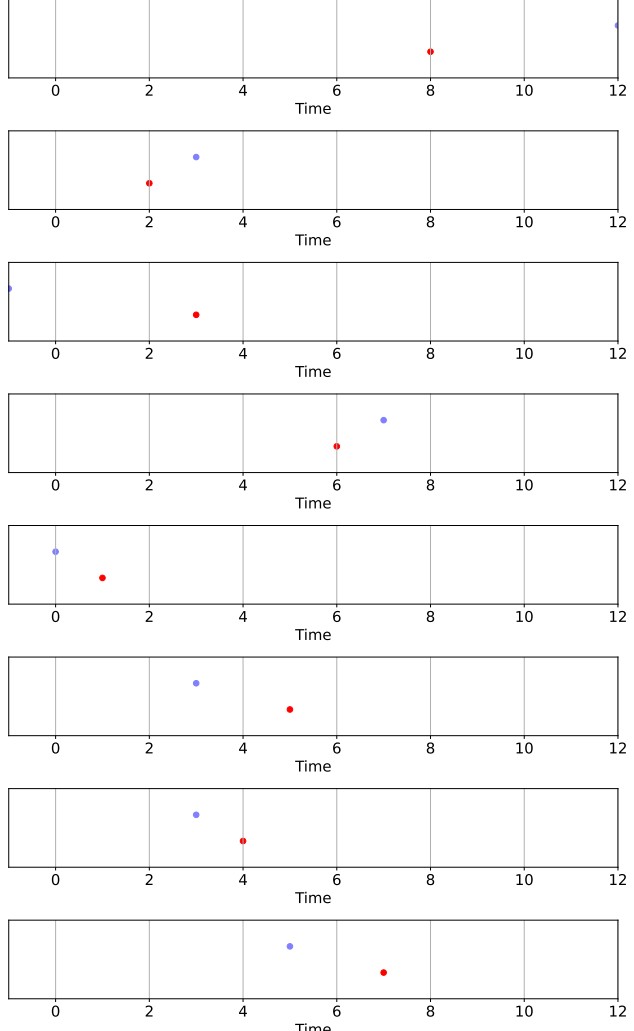

Figure 6: Example of another regenerated (encoded and decoded) sequence from `HawkesVAE` (Events Known) from NCT00003299. The blue dots denoting the specific event timestamp prediction. The red dots are the ground truth timestamps and the ground truth predictions. Each prediction is also linked with dashed lines for clarity

Figure 5 and Figure 6 show some examples of reconstructed subjects as generated by the best-performing model (`HawkesVAE` (Events Known)). Intuitively, it visually reveals that the generated data generally matches the original data.

### A.6 TRADITIONAL HAWKES PROCESS

The Hawkes Process is a double stochastic point process. The traditional Hawkes Processes assumes that past events can temporarily raise (never lower) the probability of future events, assuming that such excitation is positive, additive over the past events, and exponentially decaying with time.

$$\lambda(t) := \left( \mu + \sum_{\{t_j, t_j < t\}} e^{-\delta(t - t_j)} \right),$$

where $\delta$ is a decaying rate. Occurrence of a single event occurring at $t$ has intensity $\lambda(t)$, calculated as the sum of base intensity $\mu$ as well as all previously occurring events $t_j < t$.

### A.7 VARIATIONAL AUTOENCODER

We follow the standard formulation of VAEs (Kingma & Welling, 2013). To generate a latent variable in the multidimensional case, we first sample a noise vector $\epsilon \sim \mathcal{N}(0, \mathbf{I}$ and hidden vector $z \sim \mathcal{N}(\mu, \sigma^2 \mathbf{I})$ where $\mu$ and $\sigma^2$ are one-dimensional vectors. The relationship between the input and its latent representation is defined as: prior = $p_\theta(z)$, likelihood = $p_\theta(x|z)$, and posterior = $p_\theta(z|x)$. The intractable posterior is approximated by $q_\phi(z|x)$. We want to minimize the Kullback–Leibler divergence between $q_\phi(z|x)$ and $p_\theta(z|x)$, which in practice leads to maximizing the evidence lower bound (ELBO) for training along with the likelihood of $x$.

$$L_{\theta, \phi} = \mathbb{E}_{z \sim q_\phi(\cdot|x)}[\ln p_\theta(x|z)] - D_{KL}(q_\phi(\cdot|x)||p_\theta(\cdot)).$$

### A.8 NEURAL HAWKES PROCESS LIKELIHOOD

The following derivation is reformatted from (Mei & Eisner, 2017). For the proposed models, given complete observations of an event stream over the time interval $[0, T]$, the log-likelihood of the parameters turns out to be given by the simple formula:

$$\ell(\{(t_1, k_1), \ldots, (t_L, k_L)\}) = \sum_{j=1}^{L} \log(\lambda(t_j|\mathcal{H}_{t_j})) - \int_{t_1}^{t_L} \lambda(t|\mathcal{H}_t)dt \tag{1}$$

Where $\lambda(t) = \sum_{k=1}^{K} \lambda_k(t)$. For the rest of the proof, let us drop the $\mathcal{H}$ notation for the sake of simplicity.

The cumulative distribution function of $T_i \in (t_{i-1}, t_i)$ given history $H_{T_i}$ is given by:

$$F(t) = P(T_i \leq t) = 1 - P(T_i > t) \tag{2}$$

$P(T_i > t)$ is the probability that no events occur from $t_{i-1}$ to $t$, so we sum over the integral of those intensities.

$$F(t) = 1 - \exp\left( -\int_{t_{i-1}}^{t} \lambda(s)ds \right) \tag{3}$$

$$= 1 - \exp\left( \int_0^{t_{i-1}} \lambda(s)ds - \int_0^{t_i} \lambda(s)ds \right) \tag{4}$$

$$\tag{5}$$

The derivative can be written as:

$$f(t) = \exp\left( \int_0^{t_{i-1}} \lambda(s)ds - \int_0^{t_i} \lambda(s)ds \right) \tag{6}$$

Moreover, given the past history $\mathcal{H}_i$ and the next event time $t_i$, the type distribution of $k_i$ is given by:

$$P(k_i \mid t_i) = \frac{\lambda_{k_i}(t_i)}{\lambda(t_i)} \tag{7}$$

Therefore, we can derive the likelihood function as follows:

$$\mathcal{L} = \prod_{i:t_i \leq T} \mathcal{L}_i = \prod_{t_i \leq T} \{f(t_i)P(k_i \mid t_i)\} \tag{8}$$

$$= \prod_{i:t_i \leq T} \left\{\exp\left(\int_0^{t_{i-1}} \lambda(s)ds - \int_0^{t_i} \lambda(s)ds\right) \lambda_{k_i}(t_i)\right\} \tag{9}$$

$$\tag{10}$$

$$\tag{11}$$

Taking the log of the likelihood:

$$\ell := \log \mathcal{L} = \sum_{i:t_i \leq T} \log \lambda_{k_i}(t_i) - \sum_{i:t_i \leq T} \left(\int_0^{t_i} \lambda(s)ds - \int_0^{t_{i-1}} \lambda(s)ds\right) \tag{12}$$

$$= \sum_{i:t_i \leq T} \log \lambda_{k_i}(t_i) - \sum_{i:t_i \leq T} \left(\int_{t_{i-1}}^{t_i} \lambda(s)ds\right) \tag{13}$$

$$= \sum_{i:t_i \leq T} \log \lambda_{k_i}(t_i) - \int_{s=0}^{T} \lambda(s)ds \tag{14}$$

$$\tag{15}$$

## A.9  UTILITY / PRIVACY SPIDER PLOTS

Here, we visualize the utility/privacy trade-off that is inherent to any synthetic data generation task. Each metric is normalized for ease of visualization so that the maximum achieved metric is set as the tip of the triangle by dividing by the max. For ML Inference Privacy (where 0.5 is the ideal value), we first take the absolute value of the difference (i.e. $x = |x - 0.5|$), and then divide by the max as before.

The results are shown in Figure 7. We see a clear tradeoff, as the best-performing Distance to Closest Record model, usually VAE LSTM or PAR, performs worse on the downstream ROCAC metric. This is because the generated sequences are of poorer quality, being too different from the original. The best-performing Downstream ROCAUC models also generally have good ML Inference Privacy, which is to be expected as those models generate data that is similar to the original, which would allow for (1) better performance on the held-out test set for ROCAUC and (2) being harder to distinguish from original data.

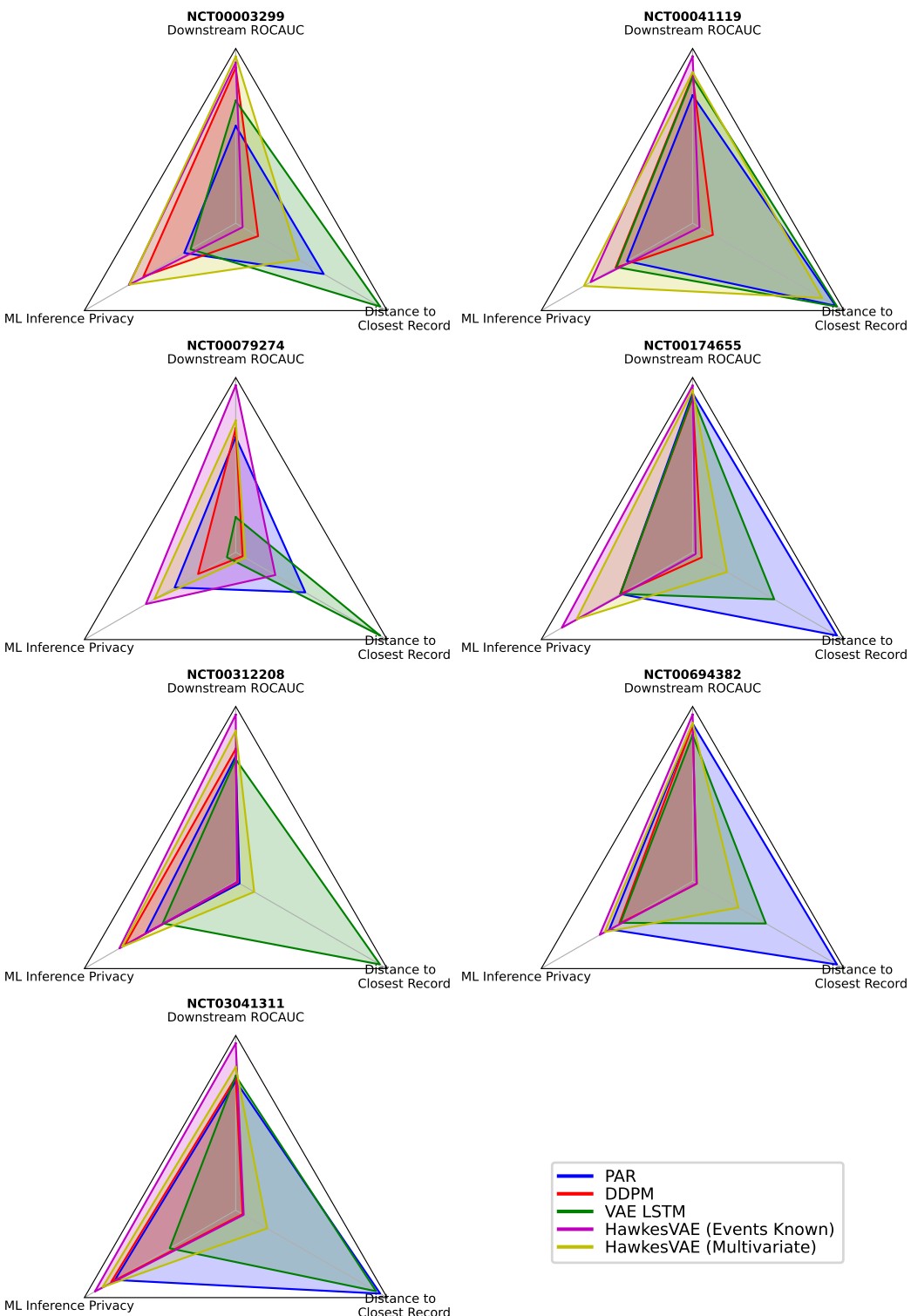

Figure 7: Spider Plots of all Models over each dataset.

### A.10 FUTURE WORK

Future work should continue exploring sequential event information. Currently, numeric values, such as lab values, are not considered. Initial experiments showed that directly regressing on the event values yielded poor results. Different methods of encoding lab values into events should be explored, such as discretization into bins. Furthermore, while we chose VAE to be the main generator framework due to its proven effectiveness, exploring Diffusion Models, which have shown superior performance over GANs as TabDDPM, is also an interesting future topic of research.

Cox-Hawkes Miscouridou et al. (2022) use a log-Gaussian Cox process (LGCP) as a prior for the background rate of the Hawkes process, allowing flexibility in capturing a wide range of underlying background effects. The computational complexity of the Hawkes process and LGCP poses challenges, but the authors introduce an efficient method for inference. Specifically, they employ a unique approach for Markov Chain Monte Carlo (MCMC) sampling, utilizing pre-trained Gaussian Process generators. This innovative technique offers direct and cost-effective access to samples during inference, addressing the computational expenses associated with the Hawkes process and LGCP.

Since Cox models are frequently used in survival analysis, a common healthcare task, this approach would be an exciting future direction to explore, potentially combining synthetic event generation with survival analysis.

