# OpenReview forum: "HawkesVAE: Sequential Patient Event Synthesis for Clinical Trials"
_ICLR.cc/2024/Conference — Submitted to ICLR 2024_

### Official Review · Reviewer_wpgs · 2023-10-28

**Soundness:** 3 good
**Presentation:** 3 good
**Contribution:** 2 fair
**Rating:** 3
**Confidence:** 4

**Summary:**

The paper proposes a VAE for modeling clinical event sequences (including both event interarrival times as well as event types) as Hawkes processes for synthetic data generation. Transformer architectures are used and the decoder samples next events based on the logits rather than greedily.

The approach is applied to clinical event sequences and compared to several baselines on downstream tasks, whether a classifier can distinguish between real and synthetic data and distance to the closest record. The results are generally favorable for the proposed method, though not always statistically significant.

Ablations are included.

**Strengths:**

1) The paper is generally well written.

2) The experiments are appropriately chosen and thorough.

3) Multiple datasets are included with details.

4) The approach appears sound.

**Weaknesses:**

1) The primary weakness is limited novelty. This is not the first paper to propose combining VAEs with Hawkes, e.g.
- Sun et al. (2021) "Towards a predictive analysis on disease progression: a variational Hawkes process model" IEEE Journal of Biomedical and Health Informatics
- Lin et al. (2021) "Disentangled deep multivariate hawkes process for learning event sequences" ICDM
- Pan et al. (2020) "A variational point process model for social event sequences" AAAI
Hence the primary point of novelty appears to be the scheme for sampling next events and the fact that using the logits is better than greedy sampling, which is not surprising.

2) Related to this is then the fact that appropriate empirical baselines may be missing since previous work combining VAEs with Hawkes processes is not included. Aside from the above papers, there are also other more closely related approaches that should be included such as Zuo et al. (2020) "Transformer Hawkes Process" ICML and Miscouridou et al. (2022) Cox-Hawkes: doubly stochastic spatiotemporal
Poisson processes TMLR.

3) Even with the limited baselines, the empirical performance is not compelling (this might be less of a concern if the contribution was more novel). While the proposed method generally has the best performance, the confidence intervals often overlap with DDPM (both on downstream performance and classification of real vs synthetic instances) - this is somewhat confusing in the presentation of the results as "best performance" is bolded for multiple methods, but competing methods with overlapping confidence intervals are not bolded.

4) While the method is well described intuitively, some specifics are not given in clear formal terms, e.g. greedy vs. logit sampling could be provided formally/mathematically, the equations for generating sequences in the encoder/decoder and combining them in the encoder should be provided.

**Questions:**

1) What do the authors consider to be the primary point of novelty over existing approaches which combine VAEs with Hawkes processes or related approaches like Transformer Hawkes?

2) Have the authors compared their approach to Transformer Hawkes Process or Cox-Hawkes?

3) Does the proposed model require the same sequence length for all patients?

---

> ### Author Response · Authors · 2023-11-19
> **Response to Reviewer wpgs**
>
> Thank you for taking the time to review our paper and your valuable feedback.
> Here are our responses to each point in order:
>
> - The primary weakness is limited novelty. This is not the first paper to propose combining VAEs with Hawkes. \ What do the authors consider to be the primary point of novelty over existing approaches which combine VAEs with Hawkes processes or related approaches like Transformer Hawkes \ Cox Hawkes?
>
> Thank you so much for pointing us to these papers; we have added them to our literature review. We believe that although our method applies VAE and Hawkes Processes together, it has merit as it outperforms synthetic tabular generation models as well as LSTMs and PAR. We also believe that the detailed level of control (allowing the user to specify event types and times) is highly valuable in settings such as clinical trial patient generation [2,3], and are not explicitly supported by other models. Finally, we have performed a literature review on the papers cited, and found that none of them, including Transformer Hawkes Models and Cox-Hawkes models, test on whole-sequence time and event generation like in our evaluations. We believe that a source of our novelty also lies in our application, where sequential event data generation is an underexplored, yet exciting field. We have not experimented with Cox-Hawkes, but this looks like an exciting future direction.
>
> - Even with the limited baselines, the empirical performance is not compelling…
>
> A quick comment regarding the performance of HawkesVAE. You are absolutely correct on the bolding of the ROC metric, and we have corrected this, as it now indicates if the original data mean ROC is within 1 standard deviation. Post initial submission, we actually realized that our downstream model was underfitting, as we discovered that our sanity check results (from training the model on original data) did not match previous existing work [1]. We have now reran the results with a more robust downstream classifier, and seen a boost in all performance for all models in general. HawkesVAE (Multivariate) outperforms the and the next best model (in 4/7 datasets and is within 1 standard deviation with the rest of the datasets). We have updated Tables 2 and 5 accordingly.
>
> - While the method is well described intuitively, some specifics are not given in clear formal terms..
>
> We agree with the reviewer, and have taken steps to clarify the variables in Section 4.1. Specifically, we clarify that the encoder model $E_(\mathcal{H}\_i) \rightarrow \hat{\mu}, \hat{\sigma}$ takes in the original event types and times, and predicts the mean and standard deviation to sample $z \sim Normal(\hat{\mu}, \hat{\sigma})$. This is used for $q_\phi(\cdot | S_z)$ in the loss function and follows the previous Transformer Hawkes Process implementation of a transformer encoder, with alternating sine and cosine waves to denote temporal information.
>
> However, the decoder model is more complicated. Hawkes Process is usually evaluated via one prediction step at a time (i.e., the input is the ground truth time-step and event type, and the task is to predict the next time step and event type). For our purposes, we need to adapt this into an autoregressive decoding scheme similar to the language modeling task.
>
> At training time, the input to the decoder $D(z, \mathcal{H}\_i) \rightarrow (\hat{t}\_{i+1}, \hat{k}_{i+1}, \lambda)$ is hidden vector $z$ and a sequence of ground truth event types and times. It is tasked with predicting the next event type $\hat{k}$, next event time $\hat{t}$, and the $\lambda$s necessary to compute $P\_\theta(\cdot|S_z)$.
>
> At inference time, the input to decoder is only $z$, and we auto-regressively decode the predicted event types and times. To predict next time and event tuple $(\hat{t}\_i, \hat{k}\_i)$, the input is the previously predicted times and events $\{(\hat{t}\_1, \hat{k}\_1), \dots, (\hat{t}\_{i-1}, \hat{k}_{i-1}))\}$). (each predicted time and event is repeatedly appended to the input).
>
> We have clarified this in section 4.1 as well as Appendix A.3.
>
> - Does the proposed model require the same sequence length for all patients?
>
> Our model supports patients of any sequence length. We have clarified this point in the manuscript (Section 5 Datasets)
> We hope that these revisions help address some of your concerns regarding the paper, and look forward to further discussion.
>
> [1] Wang, Z., Gao, C., Xiao, C., & Sun, J. (2023). AnyPredict: Foundation Model for Tabular Prediction. arXiv preprint arXiv:2305.12081.
>
> [2] Beigi, M., Shafquat, A., Mezey, J., & Aptekar, J. (2022). Simulants: Synthetic Clinical Trial Data via Subject-Level Privacy-Preserving Synthesis. In AMIA Annual Symposium Proceedings (Vol. 2022, p. 231). American Medical Informatics Association.
>
> [3] https://www.medidata.com/wp-content/uploads/2023/10/FACT-SHEET-Simulants.pdf

---

### Official Review · Reviewer_jzNy · 2023-10-31

**Soundness:** 1 poor
**Presentation:** 1 poor
**Contribution:** 1 poor
**Rating:** 3
**Confidence:** 4

**Summary:**

The paper proposes a HawkesVAE model for simulating time series (event time and type) data. HawkesVAE combines the Neural Hawkes Process (NHP) with the data encoding and decoding capabilities of Variational Autoencoders (VAEs). Experimental results show that the generated samples closely resemble the original data compared to baselines.

**Strengths:**

Learning to simulate time series (event time and type) data is an important but under-explored research topic.

**Weaknesses:**

**Novelty**
- The paper seems like a straightforward application of NHP and VAEs to the time-series data

**Clarity**
- The writing could improved to focus on the paper's key contributions. For instance, given that a previous derivation of the NHP in Section 3.2 has been proposed unless there are new insights, the NHP could be summarized.
- The description for HawkesVAE is lacking in clarity:
1) How is $p_\theta(|zS_z)$ parametized?
2)  How is $q_{\phi}(z|S_z)$  parametized?
3)  How is $z$ used in the log-likelihood $p_{\theta}(S_z|z)$?
- Figure 1:  It seems the model outputs event times and event lengths. Does the model predict event types? Why does HawkesVAE require event lengths and event types at inference time? Are these provided to baselines as well?
- Tables 2, 3, 5: HawkesVAE results are bolded even in instances where baselines are better than HawkesVAE which is misleading
-  Given that HawkesVAE requires access to real-world data to learn a generative model, the benefits of using the synthetic data over real-world data are not motivated
- How are the encoder and decoder functions specified?

**Questions:**

See above

---

> ### Author Response · Authors · 2023-11-19
> **Response to Reviewer jzNy (1/2)**
>
> Thank you for taking the time to review our paper and your valuable feedback.
>
> Here are our responses to each point in order:
>
> - The paper seems like a straightforward application of NHP and VAEs to the time-series data
>
> We believe that although our method applies two methodologies together, it has merit as it outperforms synthetic tabular generation models as well as LSTMs and PAR. We also believe that the detailed level of control (allowing the user to specify event types and times) is highly valuable in settings such as clinical trial patient generation, and are not explicitly supported by other models.
>
> - The writing could improved to focus on the paper's key contributions.
>
> This is a good point, and we have shortened the relevant work to a single page to reflect this change. We have moved the information regarding traditional Hawkes Processes and VAEs to the appendix as well.
>
> - The description for HawkesVAE is lacking in clarity / How are the encoder and decoder functions specified?
>
> We agree with the reviewer, and have taken steps to clarify the variables in Section 4.1. Specifically, we clarify that the encoder model $E_(\mathcal{H}\_i) \rightarrow \hat{\mu}, \hat{\sigma}$ takes in the original event types and times, and predicts the mean and standard deviation to sample $z \sim Normal(\hat{\mu}, \hat{\sigma})$. This is used for $q_\phi(\cdot | S_z)$ in the loss function and follows the previous Transformer Hawkes Process implementation of a transformer encoder, with alternating sine and cosine waves to denote temporal information.
>
> However, the decoder model is more complicated. Hawkes Process is usually evaluated via one prediction step at a time (i.e., the input is the ground truth time-step and event type, and the task is to predict the next time step and event type). For our purposes, we need to adapt this into an autoregressive decoding scheme similar to the language modeling task.
>
> At \textit{training time}, the input to the decoder $D(z, \mathcal{H}\_i) \rightarrow (\hat{t}\_{i+1}, \hat{k}_{i+1}, \lambda)$ is hidden vector $z$ and a sequence of ground truth event types and times. It is tasked with predicting the next event type $\hat{k}$, next event time $\hat{t}$, and the $\lambda$s necessary to compute $P\_\theta(\cdot|S_z)$.
> At \textit{inference time}, the input to decoder is only $z$, and we auto-regressively decode the predicted event types and times. To predict next time and event tuple $(\hat{t}\_i, \hat{k}\_i)$, the input is the previously predicted times and events $\{(\hat{t}\_1, \hat{k}\_1), \dots, (\hat{t}\_{i-1}, \hat{k}_{i-1}))\}$). (each predicted time and event is repeatedly appended to the input).
>
> - Figure 1: It seems the model outputs event times and event lengths. Does the model predict event types? Why does HawkesVAE require event lengths and event types at inference time? Are these provided to baselines as well?
>
> The model autoregressively predicts event types and times, and event lengths and event type information are optional inputs. The model without event type information is denoted as HawkesVAE (Multivariate), and the model with event type information is denoted as HawkesVAE (Events Known). The comparisons are indeed fair, as baseline models vs HawkesVAE (Multivariate) is given the same amount of information. We have also taken steps to clarify Figures 1 and 3.
>
> - Tables 2, 3, 5: HawkesVAE results are bolded even in instances where baselines are better than HawkesVAE which is misleading.
>
> A quick comment regarding the performance of HawkesVAE. You are absolutely correct on the bolding of the ROC metric, and we have corrected this, as it now indicates if the original data mean ROC is within 1 standard deviation. Post initial submission, we actually realized that our downstream model was underfitting, as we discovered that our sanity check results (from training the model on original data) did not match previous existing work [1]. We have now reran the results with a more robust downstream classifier, and seen a boost in all performance for all models in general. HawkesVAE (Multivariate) outperforms the and the next best model (in 4/7 datasets and is within 1 standard deviation with the rest of the datasets). We have updated Tables 2 and 5 accordingly.
>
> - Given that HawkesVAE requires access to real-world data to learn a generative model, the benefits of using the synthetic data over real-world data are not motivated
>
> We understand the reviewer’s concern regarding this. In fact, synthetic trial generation is a recently proposed field of research, and has already become an industry product. Relying on real world data to train such models is indeed an issue inherent to all synthetic data generative models, but we believe that the use cases (data insights without revealing patient personal information) are highly practical for real world applications.

---

> > ### Author Response · Authors · 2023-11-19
> > **Response to Reviewer jzNy (2/2)**
> >
> > We hope that these revisions help address some of your concerns regarding the paper, and look forward to further discussion.
> >
> > [1] Wang, Z., Gao, C., Xiao, C., & Sun, J. (2023). AnyPredict: Foundation Model for Tabular Prediction. arXiv preprint arXiv:2305.12081.
> >
> >
> > [2] Beigi, M., Shafquat, A., Mezey, J., & Aptekar, J. (2022). Simulants: Synthetic Clinical Trial Data via Subject-Level Privacy-Preserving Synthesis. In AMIA Annual Symposium Proceedings (Vol. 2022, p. 231). American Medical Informatics Association.
> >
> > [3] https://www.medidata.com/wp-content/uploads/2023/10/FACT-SHEET-Simulants.pdf

---

### Official Review · Reviewer_6QpK · 2023-11-01

**Soundness:** 3 good
**Presentation:** 3 good
**Contribution:** 2 fair
**Rating:** 5
**Confidence:** 4

**Summary:**

This paper proposed a method called HawkesVAE to combine Hawkes Process (HP) and Variational Autoencoder (VAE) for events prediction and duration estimation.

The proposed method is applied on 7 oncology trial datasets and compared with thee existing methods LSTM VAE, PARSynthesizer, and DDPM. By comparing the ROCAUC (binary classification of death), HawkesVAE tend to outperform the other three methods when number of subject is small and tend to report highter ROCAUC when original data with events/Subj rate is lowers. It’s also compared with the other three methods under the ML inference Score and the Distance to Closest Record criterion.

**Strengths:**

Generating synthetic competing risk events based on limited clinic trial data is challenging yet meaningful. This paper combined Hawkes and VAE to address sequencing events generation which is close to original data while not repeat of original data.

**Weaknesses:**

It’s not clear to me what types of events are available in those datasets, how general adverse events, serious adverse events, and death are handled and once generated would that generated event (by serious/severity) lead to more/less frequent or cancellation of afterward event generation.

**Questions:**

1.P3, What are the other events mentioned in “death event reduces the probability of most other events”?
2.P3, can the row above \lambda(t|\mathcal(H)) read as “is defined as” instead of “calculated as”? (Maybe just me, I was trying to derive this row based on previous definitions.)
3.P4, Figure 1. I didn’t quite follow how the [0,1,1,0,0] (5 events) event indicator was paired with your two event time vectors [1,3,5,6] and [1,2,3] (total 7 time points), and how matched with the event length [4,3] (two durations for the two “1” right).

---

> ### Author Response · Authors · 2023-11-19
> **Response to Reviewer 6QpK**
>
> Thank you for taking the time to review our paper and your valuable feedback.
>
> A quick comment regarding the performance of HawkesVAE. We have slightly changed our bolding, which now indicates if the original data mean ROC is within 1 standard deviation. Post initial submission, we actually realized that our downstream model was underfitting, as we discovered that our sanity check results (from training the model on original data) did not match previous existing work [4]. We have now reran the results with a more robust downstream classifier, and seen a boost in all performance for all models in general. HawkesVAE (Multivariate) outperforms the and the next best model (in 4/7 datasets and is within 1 standard deviation with the rest of the datasets). We have updated Tables 2 and 5 accordingly.
>
> Here are our responses to each point in order:
>
> - It’s not clear to me what types of events are available in those datasets, how general adverse events, serious adverse events, and death are handled and once generated would that generated event (by serious/severity) lead to more/less frequent or cancellation of afterward event generation.
>
> Each dataset contains events and the times at which they occur, e.g. medications, procedures, as well as some adverse events like vomiting, etc. We use these datasets to predict if the subject experiences the death event, which is an external label.
>
> - 1.P3, What are the other events mentioned in “death event reduces the probability of most other events”?
>
> We have changed this to be more representative of our datasets and evaluation task, as death is not actually an event that is in the training set. Rather, it is an external label of patient status, and the events consist of events and the times at which they occur, e.g. medications, procedures, as well as some adverse events like vomiting, etc. We will use these datasets to predict if the subject experiences the death event, which is an external label. We have also changed the example to “e.g., medication reduces the probability of adverse events”.
>
> - 2.P3, can the row above \lambda(t|\mathcal(H)) read as “is defined as” instead of “calculated as”? (Maybe just me, I was trying to derive this row based on previous definitions.)
>
> We have revised the text and replaced “calculated as” with “is defined as”.
>
> - 3.P4, Figure 1. I didn’t quite follow how the [0,1,1,0,0] (5 events) event indicator was paired with your two event time vectors [1,3,5,6] and [1,2,3] (total 7 time points), and how matched with the event length [4,3] (two durations for the two “1” right).
>
> We have clarified the figure so that the event lengths are labeled. Originally, I meant to demonstrate that within 5 events, events 2 and 3 occur in the subject with times [1,3,5,6] (length 4) and [1,2,3] (length 3). I realize that this is confusing and have clarified the figure with labels of a more intuitively understandable event sequence (“Vomiting” and “Fever”). Please see Figures 1 and 3.

---

### Official Review · Reviewer_S6gY · 2023-11-01

**Soundness:** 2 fair
**Presentation:** 1 poor
**Contribution:** 2 fair
**Rating:** 5
**Confidence:** 3

**Summary:**

Authors propose HawkesVAE, a method to generate synthetic event-type data. The proposed approach combines Hawkes Process Transformers with a Variational Autoencoder. The method is empirically validated on different clinical trial datasets.

**Strengths:**

- The paper addresses the challenging and important problem of generating synthetic medical time-series.
- The proposed method is interesting and sensible, combining insights from relevant methods.
- Three metrics are used to measure quality of synthetic data: real test performance after training on synthetic data, and 2 measures of similarity to real data. Although a comment on what is done in related works would be helpful!

**Weaknesses:**

Missing details and lack of clarity on the experimental investigation make it challenging to measure the added value of the proposed method (see questions in the next section).

Presentation and readability need significant improvement:
- Citations within parentheses
- “Event-type + time-gap” please replace “+” by “and”
- Contribution 3 not a sentence
- P.3 h(t)s
- P.4 “Note that the gradient of is simply”
- P.4 Use of capital $\Delta$ for differentiation is unconventional. Do authors mean the gradient operator $\nabla$, or the $\delta$ partial diff operator? I also believe authors mean that $\lambda(u)$ is differentiable wrt $u$ – not that the gradient of $\lambda$ is differentiable itself.
- P4 mulidimensional
- P4 “we get ϵ ∼ N (0,I)” – what does this refer to?
- P5 missing punctuation after “only parameterized by θ”
- P8 “and normalized timestamp, respectively.”
- P4 missing $\phi$ in KL-divergence of ELBO
- P6 Citation for CTGAN (why is it not included as a baseline?)
- Table 2: Overlap between HawkesVAE results and models trained on original data (should also be bolded).
- Too many significant figures on all tables.

**Questions:**

- Table 3: to validate this metric, what is the performance of applying the real/synthetic classifier to real data? Is it indeed 0.5? What synthetic data was this discriminator trained on?

- Is the downstream classification task just binary classification of death? What does “the LSTM and the HawkesVAE models estimate their own sequence stopping length” mean in the context of binary classification, do you also jointly regress death time? Does this mean the death event is removed from the end of training trajectories (real or synthetic)? What happens if the trajectory does not undergo any event?

- Why does training on synthetic data from Hawkes VAE occasionally beat training on the original data?

- I am not sure I understand why and how Hawkes-VAE is used for event forecasting, if it is only designed to generate synthetic data. How is this implemented in practice? Do authors sample from the posterior generated by the history up to the prediction point and then generate the rest of the trajectory? Why does giving more info (“Events Known”) result in a less accurate order?

- In what context would one expect to know/constrain the event types happening within a sequence?

- What is the error reported (CI, stderr?)?

---

> ### Author Response · Authors · 2023-11-19
> **Response to Reviewer S6gY (1/2)**
>
> Thank you for taking the time to review our paper and your valuable feedback.
>
> Thank you so much for the general writing comments, and we have clarified all sections accordingly.
>
> A quick comment regarding the performance of HawkesVAE. You are absolutely correct on the bolding of the ROC metric, and we have corrected this, as it now indicates if the original data mean ROC is within 1 standard deviation. Post initial submission, we actually realized that our downstream model was underfitting, as we discovered that our sanity check results (from training the model on original data) did not match previous existing work [4]. We have now reran the results with a more robust downstream classifier, and seen a boost in all performance for all models in general. HawkesVAE (Multivariate) outperforms the next best model (in 4/7 datasets and is within 1 standard deviation with the rest of the datasets). We have updated Tables 2 and 5 accordingly.
>
> - Why is CTGAN not included as a baseline?
>
> We believed that since TabDDPM [1] showed showed significant superior performance vs CTABGAN [2] and CTABGAN+ [3] (both of these models outperform CTGAN), we assumed that it would be a stronger baseline of general synthetic tabular generation compared to CTGAN, and left out CTGAN in favor of more recent work.
>
> - Table 3: to validate this metric, what is the performance of applying the real/synthetic classifier to real data? Is it indeed 0.5? What synthetic data was this discriminator trained on?
>
> Yes, the performance of the real-vs-synthetic classifier is indeed 0.5. The synthetic data and the real data are simply the real training data (label 0) and the generated synthetic patients from each of the methods (label 1). We have clarified this in the paper in section 5.2.
>
> - Is the downstream classification task just binary classification of death? What does “the LSTM and the HawkesVAE models estimate their own sequence stopping length” mean in the context of binary classification, do you also jointly regress death time? Does this mean the death event is removed from the end of training trajectories (real or synthetic)? What happens if the trajectory does not undergo any event?
>
> Yes, the downstream classification task is indeed binary classification of death in the patient data. Death is not an event that is explicitly in the dataset, rather, the dataset simply contains events and timestamps of medications, procedures, as well as some adverse events like vomiting. No regression is done on explicit time-to-death prediction, as our model is only trained on event prediction. Death is an external label that is also available in the dataset. We have clarified this in section 5.
>
> - Why does training on synthetic data from Hawkes VAE occasionally beat training on the original data?
>
> Occasionally, synthetic data is able to support better performance than the original dataset on downstream tasks (this behavior is also seen in TabDDPM [1]). We believe that this is due to the synthetic model generating examples that are more easily separable and/or more diverse than real data. However, this is only an hypothesis and should be investigated further in future research, but we are encouraged to see that our proposed method captures this behavior. We have added commentary on this observation in Section 5.1.
>
> - I am not sure I understand why and how Hawkes-VAE is used for event forecasting…
>
> HawkesVAE is trained via next event prediction, similar to Transformer Hawkes (To predict next time and event tuple $(t_i, k_i)$, the input is true previous times and events $\{(t_1, k_1), \dots, (t_{i-1}, k_{i-1}))\}$). Note that our main results for ML Efficiency performs this autoregressively (where each predicted time and event is repeatedly appended to the input), as we of course do not have access to the true events. Giving more info in the “Events Known” category actually surprisingly performs worse because it is actually $num_types$ different Hawkes Models, which each individually models its respective event and event times, and then combined to create a final multi-event sequence prediction. Its predictions are generally useful, but the strict ordering of the events may not be captured. This is why it performs worse in the strict forecasting sense. The general HawkesVAE model is a multivariate Hawkes process, which considers ALL events and times, so its forecasting performance should be intuitively better. We have clarified our methodology in Section 4.1 accordingly.

---

> > ### Author Response · Authors · 2023-11-19
> > **Response to Reviewer S6gY (2/2)**
> >
> > - In what context would one expect to know/constrain the event types happening within a sequence?
> >
> > This is an excellent question. In the clinical trial setting, one main concern is patient fidelity--that is--the generated patients must be significantly similar to the original patients in order for the generated data to be useful; knowing which events occur in a patient to generate a similar patient would not be unreasonable. The “Events Known” model was created to enforce ONLY simulating specific events, without consideration of all events (which may be irrelevant and confuse the generator). We have also clarified this in Section 4.1.
> >
> > - What is the error reported (CI, stderr?)?
> >
> > They are standard deviations calculated using bootstrapping, we have clarified it in Section 5.1 Utility Evaluation. Thanks for pointing this out.
> >
> > We hope that these revisions help address some of your concerns regarding the paper, and look forward to further discussion.
> >
> > [1] Kotelnikov, A., Baranchuk, D., Rubachev, I., & Babenko, A. (2023, July). Tabddpm: Modelling tabular data with diffusion models. In International Conference on Machine Learning (pp. 17564-17579). PMLR.
> >
> > [2] Zhao, Z., Kunar, A., Birke, R., & Chen, L. Y. (2021, November). Ctab-gan: Effective table data synthesizing. In Asian Conference on Machine Learning (pp. 97-112). PMLR
> >
> > [3] Zhao, Z., Kunar, A., Birke, R., & Chen, L. Y. (2022). Ctab-gan+: Enhancing tabular data synthesis. arXiv preprint arXiv:2204.00401.
> >
> > [4] Wang, Z., Gao, C., Xiao, C., & Sun, J. (2023). AnyPredict: Foundation Model for Tabular Prediction. arXiv preprint arXiv:2305.12081.

---

> > > ### Comment · Reviewer_S6gY · 2023-11-22
> > >
> > > Thank you for your response.
> > >
> > > Your new results appear promising and many of the proposed changes improve the quality of the paper. In my opinion, they represent too many changes for a rebuttal, considering most experimental results have changed. I therefore retain my score.
> > >
> > > Some additional comments/points that would benefit from further clarification:
> > > * The modelling differences between Hawkes Multivariate and Event known remain unclear to me. I now see that the latter consists of a distinct model for each type of event, but do not understand the following: "Finally, an overall prediction combines event time predictions for all known types to create a final multi-event sequence prediction." I am also unclear why authors propose to sum over the latent space of these distinct models, and how this is motivated.
> > >
> > > * I do not have access to the pre-rebuttal manuscript but see that the "Events Known" Hawkes VAE now performs better than the Multivariate version in Table 6. I may recall incorrectly, but I believe this was not the case before the rebuttal. Is there an explanation for this change in performance?
> > >
> > > Finally, please carefully proofread the manuscript as it retains presentation issues (e.g. punctuation, poor citation style).

---

> ### Author Response · Authors · 2023-11-22
> **Response to Reviewer S6gY**
>
> Thank you for your response! We would like to address each point as follows:
>
> -  too many changes for a rebuttal, considering most experimental results have changed. I therefore retain my score.
>
> The only experimental changes were made was the more robust ML efficiency evaluation, which affects Tables 2 and 5, respectively. All other changes are paper editing changes to improve writing fluency.
>
> - The modelling differences between Hawkes Multivariate and Event known remain unclear to me. I now see that the latter consists of a distinct model for each type of event, but do not understand the following: "Finally, an overall prediction combines event time predictions for all known types to create a final multi-event sequence prediction." I am also unclear why authors propose to sum over the latent space of these distinct models, and how this is motivated.
>
> Since the event type is known, we may index into the specific encoder/decoder pair as given by the event index (shown in Fig. 1). Finally, all independent event time predictions over all known types are combined via their predicted event times to create a final multi-event sequence prediction.
>
> Since the VAE model requires a single embedding to sample a latent vector, we sum the patient embedding output of all expert event encoders, and pass this joint embedding to the expert decoders. The decoder is trained to generate the predicted time and type sequence of its respective expert event.
>
> Essentially, this is purely motivated by our limitation of having a fully generative model. We have to constrain the model to a constant-size embedding space, as it would be unfair for to have varying size latent spaces for each patient.
>
> We have clarified this in section 4.1 Event Type Information (page 5).
>
> - "Events Known" Hawkes VAE now performs better than the Multivariate version in Table 6. I may recall incorrectly, but I believe this was not the case before the rebuttal. Is there an explanation for this change in performance?
>
> The performance of these 2 models remains the same as in the pre-rebuttal version. We have only removed the bolding of the table text, as we believed it was not clear. “Events known” has access to the ground truth Event Types Information, and it thus an unfair comparison. We have decided to separate this using a dividing line instead.
>
> - Finally, please carefully proofread the manuscript as it retains presentation issues (e.g. punctuation, poor citation style).
>
> We have made many grammatical edits in the paper, as well as highlighted our citations for improved visibility. If there are any further major grammatical errors, we would be happy to address them.

---

### Author Response · Authors · 2023-11-19
**General Response: New manuscript revision**

We would like to thank all reviewers for taking the time to review and make comments to improve our paper. As an overall summary, we have:

1. Made extensive changes to the paper (highlighted in red clarifying sections per reviewer comments)
2. Updated the Main Figure 1 for clarity
3. Updated the description in section 4.1 with more mathematical precision,
4. Updated experiments Table 2 and 5 with a more robust evaluation model and improved bolding).

---

> ### Author Response · Authors · 2023-11-22
> **To all Reviwers**
>
> As the discussion period is coming to an end tomorrow, we kindly ask the reviewers to review our response to your comments and let us know if there are any further queries. Alternatively, per the paper revisions, if a consideration for raising the score of the paper could be made, we would be extremely grateful. We eagerly anticipate your comments and are committed to addressing any remaining concerns before the discussion period concludes.

---

### Meta-Review · Area_Chair_BhFa · 2023-12-10

**Metareview:**

The authors propose HawkesVAE, deep generative model to generate synthetic data from complex Hawkes processes in the context of medical time-series data. As acknowledged by the reviewers, the paper focuses on an important application of VAEs, with a sensible methodology for the problem at the hand. The main weakness of the paper lies on the fact that in its original submitted form, it was clearly  not ready for publication. I acknowledge that the authors have significantly clarify some of the major issues with the paper during the rebuttal, but I also agree with reviewer S6gY, that the changes in the paper are significant and thus would require another full round of detailed reviews prior to publication. This is a necessary step to assess i) if the major points of concern, which include among others motivation of the proposed method, description of the experiments, discussion on previous approaches; and ultimately, ii)  the main contributions of the paper.

**Justification For Why Not Higher Score:**

This is a clear paper that was under the ICLR bar, and while authors have addressed many of the issued pointed out by the reviewers, I believe that the changes are major enough to require for another complete round of reviews.

**Justification For Why Not Lower Score:**

N/A

---

### Decision · Program_Chairs · 2024-01-16

Reject